

# Individualized responses to velocity-based *versus* percentage-based resistance training in combat sports athletes: the influence of baseline characteristics

JiaYong Chen[1,*], Beiwang Deng[1,*], Tianyuan He[1], Jiaxin He[1], Duanying Li[1], Min Lu[2] and Jian Sun[1,3,4]

[1] School of Athletic Training, Guangzhou Sport University, Guangzhou, China
[2] Wushu School, Guangzhou Sport University, Guangzhou, China
[3] Guangdong Provincial Key Laboratory of Human Sports Performance Science, Guangzhou, China
[4] Badminton Technique and Tactics Analysis and Diagnosis Laboratory, Guangzhou Sports University, Guangzhou, China
[*] These authors contributed equally to this work.

Corresponding authors
Min Lu, lm3899@sina.com
Jian Sun, sunjian@gzsport.edu.cn

## ABSTRACT

**Background**. Traditional percentage-based resistance training (PBRT) is a cornerstone of strength development, but its fixed nature may not account for daily fluctuations in athlete readiness. Velocity-based resistance training (VBRT) has been proposed as a superior alternative for power development due to its auto-regulatory capabilities, but its efficacy in highly trained combat sports athletes remains contested.

**Objective**. This study aimed to compare the effects of velocity-based resistance training (VBRT) and percentage-based resistance training (PBRT) on upper limb strength, general power, and sport-specific power in combat sports athletes, and to explore individualized training responses.

**Methods**. A randomized parallel-group controlled trial was conducted, recruiting 24 male university combat sports athletes (age: $21.5 \pm 2.1$ years; training experience: $4.8 \pm 1.5$ years; baseline bench press one-repetition maximum (1RM): $95.4 \pm 10.2$ kg). Participants were randomly assigned to either a VBRT group ($n = 12$) or a PBRT group ($n = 12$) for an 8-week bench press intervention. Pre- and post-intervention tests included bench press 1RM, medicine ball throws, and power in the Seoi-nage (shoulder throw).

**Results**. A $2 \times 2$ mixed-model analysis of variance (ANOVA) revealed that while both groups significantly improved in most metrics, the PBRT group demonstrated significantly greater improvements in several key areas. Significant Group × Time interactions were found favoring the PBRT group for 4kg medicine ball velocity ($p < .001$), power ($p = .012$), and distance ($p = .020$), as well as for sport-specific power in both the left ($p < .001$) and right ($p = .018$) Seoi-nage. Crucially, at post-test, the PBRT group's left Seoi-nage power was significantly higher than the VBRT group's ($p = .035$). Exploratory cluster analysis identified three distinct athlete subgroups, and PBRT elicited superior or comparable training adaptations across all of them.

**Conclusion**. For the cohort of university-level combat sports athletes in this study, PBRT was a more effective training methodology than VBRT for enhancing both general and sport-specific power. These findings challenge the assumption of VBRT's universal

superiority for power development and highlight the continued efficacy and robustness of traditional PBRT for strength and conditioning in this population.

# INTRODUCTION

Judo and wrestling are high-intensity combat sports demanding exceptional levels of strength and power (*Franchini et al., 2011*). Upper limb strength is crucial for executing key techniques like throws and takedowns, which are determinantal to competitive success (*Franchini et al., 2011*; *García-Pallarés et al., 2011*). Studies have consistently shown that elite combat sports athletes possess significantly greater upper-body strength and power compared to their sub-elite counterparts (*Marques et al., 2019*; *Almeida et al., 2021*), underscoring the importance of optimizing resistance training programs.

Traditionally, strength training protocols have been prescribed using percentage-based resistance training (PBRT), where loads are determined as a percentage of an athlete's one-repetition maximum (1RM) (*Almeida et al., 2021*). However, a primary criticism of PBRT is its rigidity, as it does not inherently account for daily fluctuations in an athlete's readiness caused by factors like sleep, nutrition, and psychological stress (*Jones et al., 2019*; *Harris, Foulds & Latella, 2019*; *Grgic et al., 2018*). Despite these limitations, PBRT has remained a cornerstone of strength and conditioning for decades due to its proven efficacy in delivering a potent and structured stimulus for strength development (*National Strength & Conditioning Association (US), 2016*).

In response to the limitations of PBRT, velocity-based training (VBT) has emerged as a popular alternative. VBT uses movement velocity as a primary metric to monitor and regulate training intensity in real-time. This study investigates a specific application, velocity-based resistance training (VBRT), which involves adjusting the external load (resistance) based on real-time velocity feedback to maintain a target velocity zone (*Balsalobre-Fernández & Torres-Ronda, 2021*). The strong correlation between movement velocity and relative intensity (%1RM) (*González-Badillo & Sánchez-Medina, 2010*) allows VBRT to auto-regulate the training load. It is often hypothesized that this real-time feedback enhances neuromuscular adaptations and is superior for power development (*Dorrell, Smith & Gee, 2020*). However, the application of VBRT in highly trained populations has yielded inconsistent findings, with some meta-analyses showing no clear advantage over traditional PBRT (*Orange et al., 2022*; *Liao et al., 2021*).

Considering the unique demands of combat sports (*Da Silva et al., 2021*), the efficacy of VBRT remains contested. It is plausible that the consistent, high-intensity stimulus of PBRT may provide a more robust stimulus for adaptation. The high degree of mechanical tension induced by completing all prescribed sets and reps with a fixed heavy load is a known primary driver of strength adaptation (*Schoenfeld, 2010*). Therefore, this study aimed to

compare the effects of 8 weeks of VBRT and PBRT on upper limb maximal strength, general power, and sport-specific power in university-level male judo and wrestling athletes.

Specifically, we addressed the following questions: (1) which training method is more effective at improving strength and power? (2) Does one method show superior transfer to a sport-specific technique like the Seoi-nage? (3) Can we identify subgroups of athletes who respond differently to these training modalities? Contrary to the prevailing VBRT-superiority hypothesis, we hypothesized that the structured, high-effort nature of PBRT would elicit comparable or even superior adaptations in this athletic population.

## METHODS

### Experimental design

This study employed a randomized parallel-group controlled trial design to investigate the effects of an 8-week VBRT *versus* PBRT intervention on upper limb maximal strength (bench press 1RM), general power (average velocity, average power, and distance in four kg and six kg medicine ball throws), and sport-specific power (average power of left and right Seoi-nage) in combat sports athletes. The study also aimed to analyze individual differences in training responses.

The experiment lasted 11 weeks, comprising a 1-week familiarization period, a 1-week baseline testing phase (T0), an 8-week training intervention, and a 1-week post-testing phase (T1). During the familiarization and baseline testing, participants were randomly assigned to groups using a random number generator. Prior to randomization, all participants were numbered, and a random sequence was generated. Participants were then assigned to either the VBRT group ($n = 12$) or the PBRT group ($n = 12$) according to the sequence. The group allocation process is illustrated in Fig. 1.

Bench press 1RM, medicine ball throw tests, and Seoi-nage tests were conducted before (T0) and after (T1) the training intervention. To minimize the influence of daily performance fluctuations (*Bishop, Jones & Woods, 2008*) all testing sessions were conducted following a 48-hour rest period, at the same time of day, and under similar environmental conditions. The experimental design is depicted in Fig. 2.

### Participants

To ensure the validity of the study, G*Power 3.1 software (*Faul et al., 2007*) was used to calculate the required sample size. With an effect size set at 0.5, an alpha level at 0.05, and a statistical power of 0.95, the calculation indicated a minimum requirement of 16 participants. Considering potential attrition, 24 male university combat sports athletes were recruited.

Athletes met the following inclusion criteria: (1) ≥3 years of specialized training in wrestling or judo, with competitive levels deemed homogeneous (all were university-level competitors); (2) age over 18 years; (3) no musculoskeletal injuries in the past year; and (4) at least two years of consistent resistance training experience, with proficiency in the bench press technique confirmed during the familiarization week. All 24 participants completed all planned training sessions, resulting in 100% training adherence. The PBRT group consisted of seven judo and five wrestling athletes, while the VBRT group consisted of six

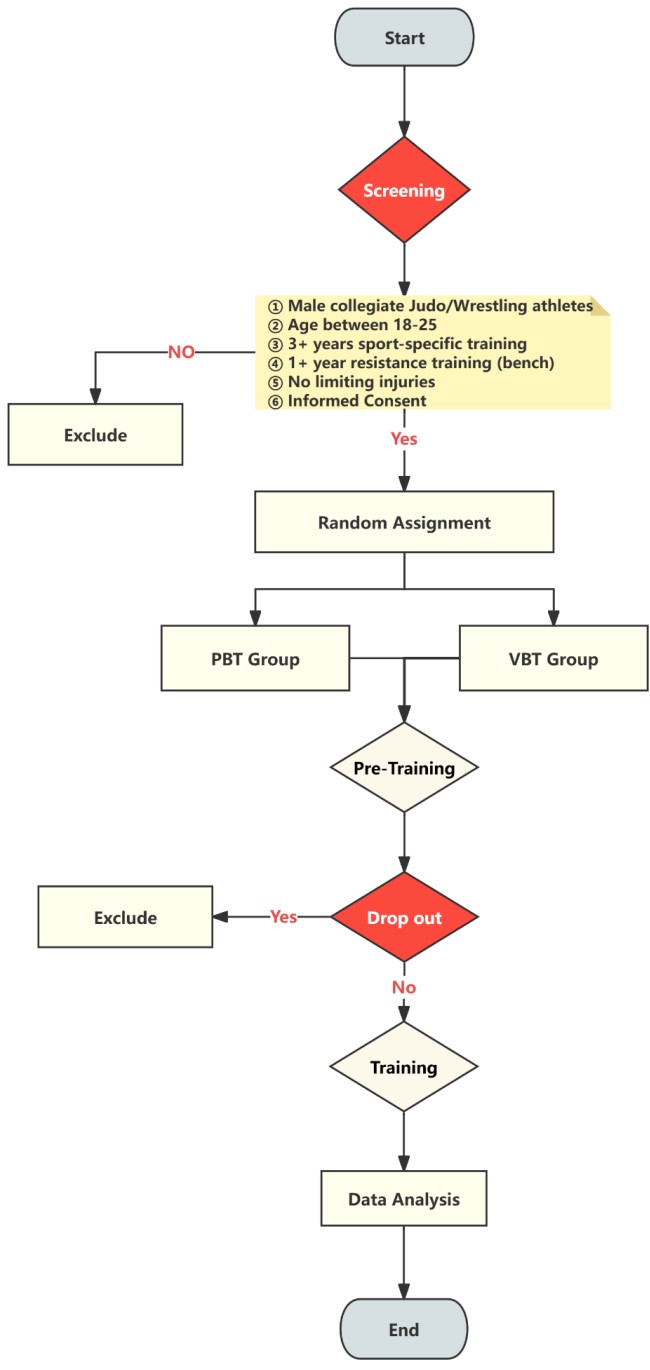

**Figure 1** Flow chart of experimental grouping.

judo and six wrestling athletes; a chi-square test confirmed no significant difference in the distribution of sports between groups ($\chi^2(1) = 0.168$, $p = 0.682$). Participant baseline information is presented in Table 1.

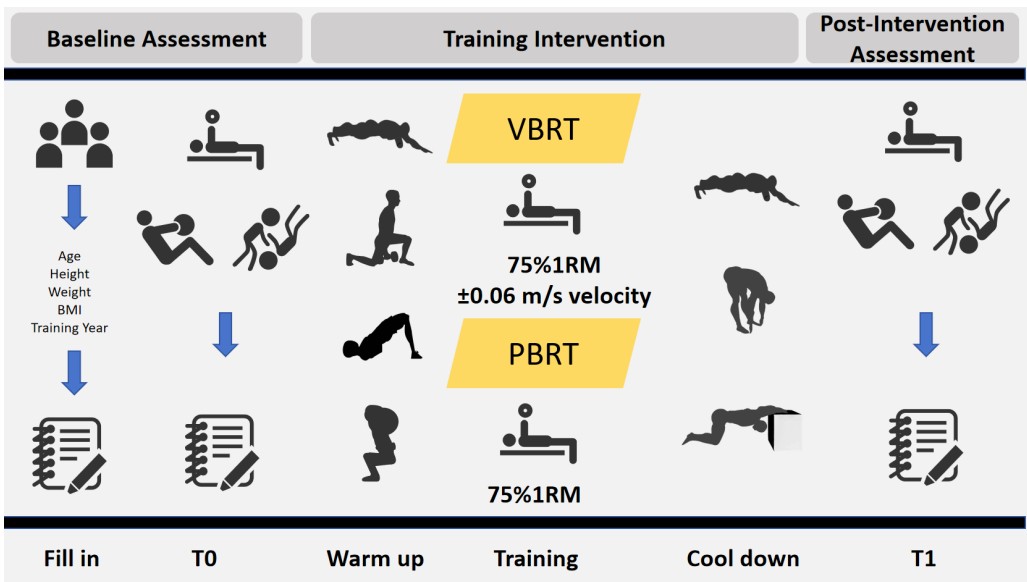

**Figure 2** Experimental design.

**Table 1 Participant baseline information.**

| Baseline information | Group | |
| --- | --- | --- |
| | (n = 12) | (n = 12) |
| Age (years) | 21.83 ± 0.58 | 20.17 ± 1.03 |
| Height (cm) | 173.92 ± 4.29 | 174.58 ± 5.37 |
| Weight (kg) | 65.77 ± 5.75 | 68.38 ± 7.77 |
| Training experience (years) | 7.58 ± 2.97 | 5.5 ± 2.02 |

This study adhered to the ethical principles outlined in the Declaration of Helsinki (*World Medical Association, 2013*). All participants provided written informed consent after being informed of the risks and benefits associated with the study. The study was approved by the Institutional Ethics Committee (Approval No. 2023LCLL-73).

## Testing procedures

To provide individualized training parameters for the VBRT group, all participants underwent an individualized load-velocity profile (LVP) construction during the baseline assessment. This study employed a multi-set, incrementally loaded protocol with four-six ascending loads, a method consistent with recommended best practices for establishing reliable load-velocity relationships (*Balsalobre-Fernández & Torres-Ronda, 2021*). The specific testing protocol is detailed in Table 2. A GymAware linear position transducer (Kinetic Performance Technology, Canberra, Australia) (*Orange et al., 2020*) was used to measure mean concentric velocity (MCV).

The primary outcome measures of this study are detailed in Table 3. The medicine ball throw was selected as a primary measure of upper-body power due to its established validity
**Table 2  LVP testing protocol.**

| Group | Repetitions | Intensity |
|---|---|---|
| 1 | 2–6 | 20–40% 1RM |
| 2 | 2–4 | 40–50% 1RM |
| 3 | 1–3 | 60–70% 1RM |
| 4 | 1–2 | 70–80% 1RM |
| 5 | 1 | >80% 1RM |

**Table 3  Test Indicaters.**

| Category | Indicator | Description | Unit |
|---|---|---|---|
| Basic strength | Bench press 1RM | Bench press test using a standard barbell (*Cohen, 2009*), recording the maximum weight the athlete can successfully lift once. | kg |
| Upper limb power | Explosive medicine ball throw | Athletes lie supine on a bench, use a Push band to monitor explosiveness, and perform explosive horizontal medicine ball throws with a 4/6 kg medicine ball in both hands (*Banyard et al., 2021*), recording average velocity, average power, and best distance measured by tape measure. The same specification of medicine ball was used for testing, and researchers reminded athletes to maintain the same starting posture and use a uniform exertion method before each test. | m/s, Watt, meters |
| | 75% 1RM bench press velocity | Athletes use the GymAware linear position transducer real-time feedback system (*Willardson, 2007*) to perform bench press at 75% 1RM load, recording the average concentric velocity of 3 valid repetitions (*National Strength & Conditioning Association (US), 2016*; *Schoenfeld, 2010*, *Behm & Sale, 1993*; *Behm et al., 2024*). | m/s |
| Sport-specific power | Seoi-nage | Athletes use the Versa Pulley (specific model to be added) variable resistance training device, choose a left or right stance according to their actual combat posture, and perform explosive power tests simulating the complete Seoi-nage technique. During testing, Versa Pulley records the maximum power output of each tester. The best score of average power on both sides is selected as the test result. During the test, ensure that the starting posture, rope pulling speed, and amplitude remain consistent. | m/s, Watt, meters |

and reliability (*Strand et al., 2023*; *Clemons, Campbell & Jeansonne, 2010*). The four kg and six kg loads were selected to provide a comprehensive assessment of the force-velocity spectrum, as the load-velocity relationship in medicine ball throws is a valid indicator of the upper-body's maximal capacities to produce force and velocity (*Marovic et al., 2025*).

## Experimental protocol

All participants underwent an eight-week training intervention, twice per week, performing the bench press exercise. The bench press was selected as it is a fundamental measure of upper-body strength (*National Strength & Conditioning Association (US), 2016*) and has

**Table 4 Experimental protocol.**

| Section | Exercise | Sets × Reps | Rest | Load | |
|---|---|---|---|---|---|
| | | | | VBRT | PBRT |
| | Fascial mobilization | 8 × 1 | / | / | / |
| Preparation phrase | Dynamic stretching | 1 × 8 | 20 s | / | / |
| | Stability activation | 2 × 8 | 30 s | / | / |
| | Movement integration | 2 × 8 | 30 s | 4 kg Medicine ball | |
| Main phrase | Bench press | 4 × 6 | 120 s | 75% 1RM ±0.06 m/s | 75% 1RM |
| Recovery phrase | | | Stretching | | |

**Notes.**

Bench press velocities under different loads were used to construct LVP during the baseline testing week. Target velocity ranges were set based on the velocity value corresponding to 75% 1RM on the LVP. During the first training session, the VBRT group used the estimated 75% 1RM as the starting load and fine-tuned it based on the actual velocity performance of the first set to better approximate the individualized 75% 1RM level.

been shown to correlate with performance in elite combat athletes (*Marques et al., 2019*; *Almeida et al., 2021*).

Each training session consisted of four sets of six repetitions with a target relative load of 75% 1RM and 120 s of rest between sets. All athletes performed a standardized warm-up before training and a cool-down of 10 min of static stretching after training. During all sessions, a researcher provided technical supervision, safety spotting, and consistent verbal encouragement (*Nagata et al., 2020*; *Weakley et al., 2020*). The specific experimental protocol is shown in Table 4.

The VBRT group's training load was auto-regulated based on the individualized LVP corresponding to 75% 1RM (*Balsalobre-Fernández & Torres-Ronda, 2021*). When the average concentric velocity for a given set deviated by more than ±0.06 m/s from the target velocity, the load was adjusted for the next set. This threshold was selected based on previous literature (*Pareja-Blanco et al., 2020*; *Suchomel et al., 2021*) as it represents a meaningful change in performance. If the average concentric velocity was more than 0.06 m/s below the target, the load was reduced by 2.5 kg; conversely, if it was more than 0.06 m/s above the target, the load was increased by 2.5 kg.

The PBRT group employed a traditional percentage-based resistance training method. Participants trained with a fixed load equivalent to 75% of their most recently tested 1RM for all sets and repetitions (*Liao et al., 2021*).

## Data analysis

Data were analyzed using R (version 4.4.3) with the tidyverse, rstatix, factoextra, and ggstatsplot packages. The significance level for all statistical tests was set at $\alpha = 0.05$.

A 2 (Group: VBRT, PBRT) × 2 (Time: pre-test, post-test) mixed-model analysis of variance (ANOVA) was conducted for each dependent variable to assess the primary intervention effects. The primary outcome of interest was the Group × Time interaction. If a significant interaction was found, post-hoc tests with Bonferroni correction were performed. Effect sizes were calculated using partial eta-squared ($\eta p^2$) for ANOVA and Cohen's d for *t*-tests (*Cohen, 2009*).

**Table 5  Pre- and post-intervention performance data.**

| | VBRT group ($n = 12$) | | PBRT group ($n = 12$) | |
|---|---|---|---|---|
| Performance indicator | Pre-Test | Post-Test | Pre-Test | Post-Test |
| backthrow_left | 309.75 ± 55.37 | 343.08 ± 56.71 | 320.67 ± 50.36 | 391.33 ± 48.12 |
| backthrow_right | 308.08 ± 65.43 | 350.42 ± 67.46 | 323.58 ± 44.41 | 392.33 ± 50.66 |
| bench | 63.33 ± 8.88 | 74.58 ± 11.37 | 59.58 ± 7.53 | 74.38 ± 11.19 |
| push4_dis | 4.34 ± 0.53 | 4.56 ± 0.49 | 3.99 ± 0.37 | 4.47 ± 0.52 |
| push4_power | 162.16 ± 61.46 | 173.66 ± 62.34 | 132.49 ± 31.14 | 186.65 ± 52.09 |
| push4_speed | 1.26 ± 0.32 | 1.3 ± 0.3 | 1.14 ± 0.18 | 1.43 ± 0.22 |
| push6_dis | 3.92 ± 0.46 | 4.12 ± 0.44 | 3.58 ± 0.37 | 3.89 ± 0.46 |
| push6_power | 199.42 ± 62.59 | 221.06 ± 66.02 | 165.29 ± 50.14 | 204.72 ± 52.03 |
| push6_speed | 1.24 ± 0.3 | 1.33 ± 0.25 | 1.11 ± 0.21 | 1.42 ± 0.27 |

**Notes.**

Independent samples $t$-test or Mann–Whitney $U$ test. No significant between-group differences were found at baseline (pre-test) for any indicator (all $p > 0.05$).

To further investigate factors influencing training outcomes, two exploratory analyses were conducted.

- **Regression analysis:** to explore the influence of baseline levels on training adaptations, a composite baseline performance indicator was first created for each athlete by standardizing ($z$-score) and averaging their scores across all baseline tests. Linear regression models were then constructed to analyze the relationship between this composite baseline indicator, the training group (VBRT $vs.$ PBRT), and their interaction on the relative change (%) of key outcome measures (*e.g.*, bench press 1RM, left Seoi-nage power).

- **Cluster analysis:** to identify potential athlete subgroups, a K-means cluster analysis was performed on the standardized baseline performance data. The elbow method was used to determine the optimal number of clusters (K). Principal component analysis (PCA) was subsequently used to visualize the identified clusters. This analysis aimed to determine if distinct athlete profiles exhibited different response patterns to the VBRT and PBRT interventions.

## RESULTS

### Baseline characteristics and training adherence

All 24 participants completed the 8-week intervention with 100% adherence to their respective training protocols. At baseline (pre-test), independent samples $t$-tests confirmed that there were no significant between-group differences for any demographic or performance measure, including bench press 1RM, all medicine ball throw indicators, and sport-specific power in the Seoi-nage (all $p > 0.05$). This indicates that the randomization process resulted in two comparable groups prior to the intervention (Table 5).

### Main intervention effects: strength and power adaptations

To determine the effects of the different training protocols, a series of 2 (Group: VBRT, PBRT) × 2 (Time: pre-test, post-test) mixed-model ANOVAs were performed on each dependent variable. The primary findings are summarized in Table 6, with detailed results for key variables presented below.

**Table 6  Summary of mixed-model ANOVA results.**

| Effect | DFn | DFd | F | p | $\eta p^2$ |
|---|---|---|---|---|---|
| bench | | | | | |
| group | 1 | 22 | 0.261 | 0.614 | 0.012 |
| test_time | 1 | 22 | 135.228 | <.001 | 0.860 |
| group:test_time | 1 | 22 | 2.501 | 0.128 | 0.102 |
| push4_speed | | | | | |
| group | 1 | 22 | 0.010 | 0.923 | 0.000 |
| test_time | 1 | 22 | 42.946 | <.001 | 0.661 |
| group:test_time | 1 | 22 | 24.480 | <.001 | 0.527 |
| push4_power | | | | | |
| group | 1 | 22 | 0.169 | 0.685 | 0.008 |
| test_time | 1 | 22 | 17.945 | <.001 | 0.449 |
| group:test_time | 1 | 22 | 7.570 | 0.012 | 0.256 |
| push4_dis | | | | | |
| group | 1 | 22 | 1.301 | 0.266 | 0.056 |
| test_time | 1 | 22 | 43.313 | <.001 | 0.663 |
| group:test_time | 1 | 22 | 6.286 | 0.02 | 0.222 |
| push6_speed | | | | | |
| group | 1 | 22 | 0.050 | 0.826 | 0.002 |
| test_time | 1 | 22 | 20.715 | <.001 | 0.485 |
| group:test_time | 1 | 22 | 6.548 | 0.018 | 0.229 |
| push6_power | | | | | |
| group | 1 | 22 | 1.358 | 0.256 | 0.058 |
| test_time | 1 | 22 | 9.953 | 0.005 | 0.311 |
| group:test_time | 1 | 22 | 0.844 | 0.368 | 0.037 |
| push6_dis | | | | | |
| group | 1 | 22 | 2.891 | 0.103 | 0.116 |
| test_time | 1 | 22 | 23.235 | <.001 | 0.514 |
| group:test_time | 1 | 22 | 0.989 | 0.331 | 0.043 |
| backthrow_left | | | | | |
| group | 1 | 22 | 1.969 | 0.175 | 0.082 |
| test_time | 1 | 22 | 139.189 | <.001 | 0.864 |
| group:test_time | 1 | 22 | 17.936 | <.001 | 0.449 |
| backthrow_right | | | | | |
| group | 1 | 22 | 1.554 | 0.226 | 0.066 |
| test_time | 1 | 22 | 114.989 | <.001 | 0.839 |
| group:test_time | 1 | 22 | 6.503 | 0.018 | 0.228 |

For maximal strength, the analysis of bench press 1RM revealed no significant Group × Time interaction (F(1, 22) = 2.50, $p = 0.128$, $\eta p^2 = 0.102$). This indicates that the magnitude of strength improvement did not differ significantly between the VBRT and PBRT groups over the 8-week period. However, a large and significant main effect of Time was observed (F(1, 22) = 135.23, $p < 0.001$, $\eta p^2 = 0.860$), demonstrating that both

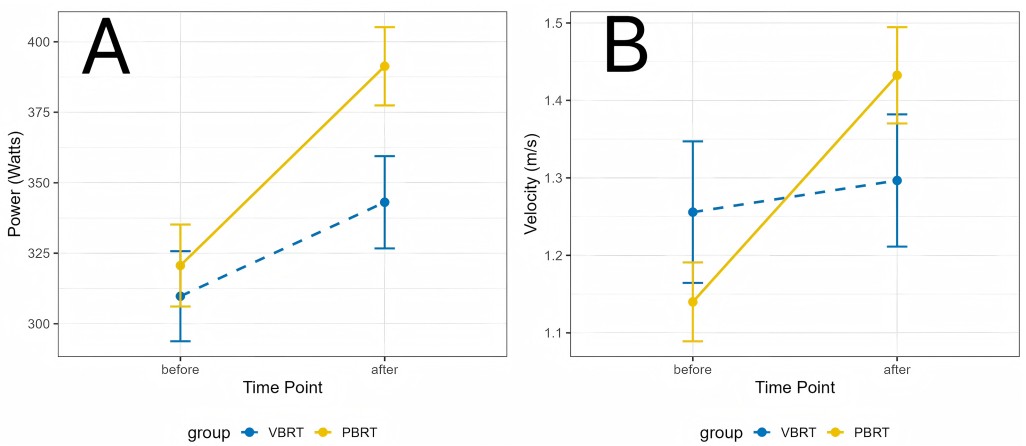

**Figure 3 Training-induced changes in key performance indicators.**

training methods were highly effective at increasing maximal upper limb strength from pre- to post-test.

In contrast to maximal strength, the analyses of general power revealed a consistent pattern of superior adaptation in the PBRT group. For the four kg medicine ball speed, a large and significant Group × Time interaction was found ($F(1, 22) = 24.48$, $p < 0.001$, $\eta p^2 = 0.527$). Post-hoc analysis was conducted to decompose this interaction. It revealed that while the PBRT group experienced a significant increase in velocity from pre- to Post-test ($p < 0.001$), the change within the VBRT group was not statistically significant ($p = 0.099$). This differential response is visually represented in Fig. 3B.

This pattern of PBRT-led superiority was replicated across other power metrics. Significant Group × Time interactions were also found for four kg medicine ball power ($F(1, 22) = 7.57$, $p = 0.012$, $\eta p^2 = 0.256$), four kg medicine ball distance ($F(1, 22) = 6.29$, $p = 0.020$, $\eta p^2 = 0.222$), and 6 kg medicine ball speed ($F(1, 22) = 6.55$, $p = 0.018$, $\eta p^2 = 0.229$). In each case, the interaction was driven by a more pronounced improvement in the PBRT group. For the remaining six kg medicine ball power and distance measures, no significant Group × Time interactions were observed ($p > 0.05$).

The advantage of the PBRT protocol extended to the sport-specific power assessments. For left Seoi-nage power, a strong Group × Time interaction was observed ($F(1, 22) = 17.94$, $p < 0.001$, $\eta p^2 = 0.449$). As illustrated in Fig. 3A, both groups significantly improved from pre- to post-test, but the magnitude of improvement was substantially larger in the PBRT group. Crucially, the post-hoc comparison of post-test values revealed that the PBRT group demonstrated significantly higher left Seoi-nage power than the VBRT group ($p = 0.035$).

A similar significant interaction was found for right Seoi-nage power ($F(1, 22) = 6.50$, $p = 0.018$, $\eta p^2 = 0.228$). Although both groups improved significantly, the pattern again indicated a more favorable adaptation for the PBRT group. However, the direct between-group comparison at Post-test did not reach statistical significance ($p = 0.100$).

**Exploratory analysis of individual differences**

To further investigate the factors influencing these training outcomes, exploratory regression and cluster analyses were performed.

Linear regression models were constructed to determine if an athlete's composite baseline performance level predicted their relative improvement (% change) and whether this relationship differed between groups.

For the relative change in bench press 1RM, the model did not reach overall significance ($F(3, 20) = 2.54$, $p = 0.085$). However, a significant main effect for the training group emerged ($p = 0.015$), indicating that the PBRT group's percentage improvement was significantly greater than the VBRT group's, even after accounting for baseline levels.

For the relative change in left Seoi-nage power, the overall model was statistically significant ($F(3, 20) = 3.92$, $p = 0.024$). Again, a significant main effect for the group was found ($p = 0.013$), confirming that the PBRT group achieved a greater percentage improvement in sport-specific power. In both models, neither the composite baseline level itself nor its interaction with the group was a significant predictor of training gains ($p > 0.05$).

- To explore whether distinct athlete profiles exhibited different training responses, a K-means cluster analysis was performed on the standardized baseline data. This process identified three distinct subgroups (Fig. 4): Cluster 1: "Speed-Dominant" ($n = 6$), characterized by the highest medicine ball velocities and power outputs but moderate strength.
- Cluster 2: "Underdeveloped" ($n = 8$), exhibiting the lowest performance across most power and sport-specific metrics.
- Cluster 3: "Technique-Dominant" ($n = 10$), demonstrating the highest sport-specific Seoi-nage power but the lowest baseline bench press strength.

An analysis of the training response within these clusters revealed a consistent trend. As shown in Fig. 5, the PBRT group achieved greater average improvements in both bench press 1RM and left Seoi-nage power across all three subgroups. This pattern was particularly pronounced for the "Underdeveloped" and "Technique-Dominant" clusters, where the gains in the PBRT group substantially outpaced those in the VBRT group. Notably, even the "Speed-Dominant" athletes, who might have been hypothesized to benefit most from VBRT, also demonstrated superior or comparable adaptations under the PBRT protocol.

## DISCUSSION

The purpose of this study was to compare the effects of VBRT and PBRT on upper limb strength and power in combat sports athletes and to explore individualized training responses. Contrary to our initial hypothesis and much of the prevailing literature that champions VBRT for power development (*Banyard et al., 2021*), our findings indicate that PBRT was a more effective training modality for this cohort of university-level combat athletes. Specifically, PBRT led to significantly greater improvements in measures of general upper-body power and sport-specific power, challenging the notion of VBRT's universal superiority.

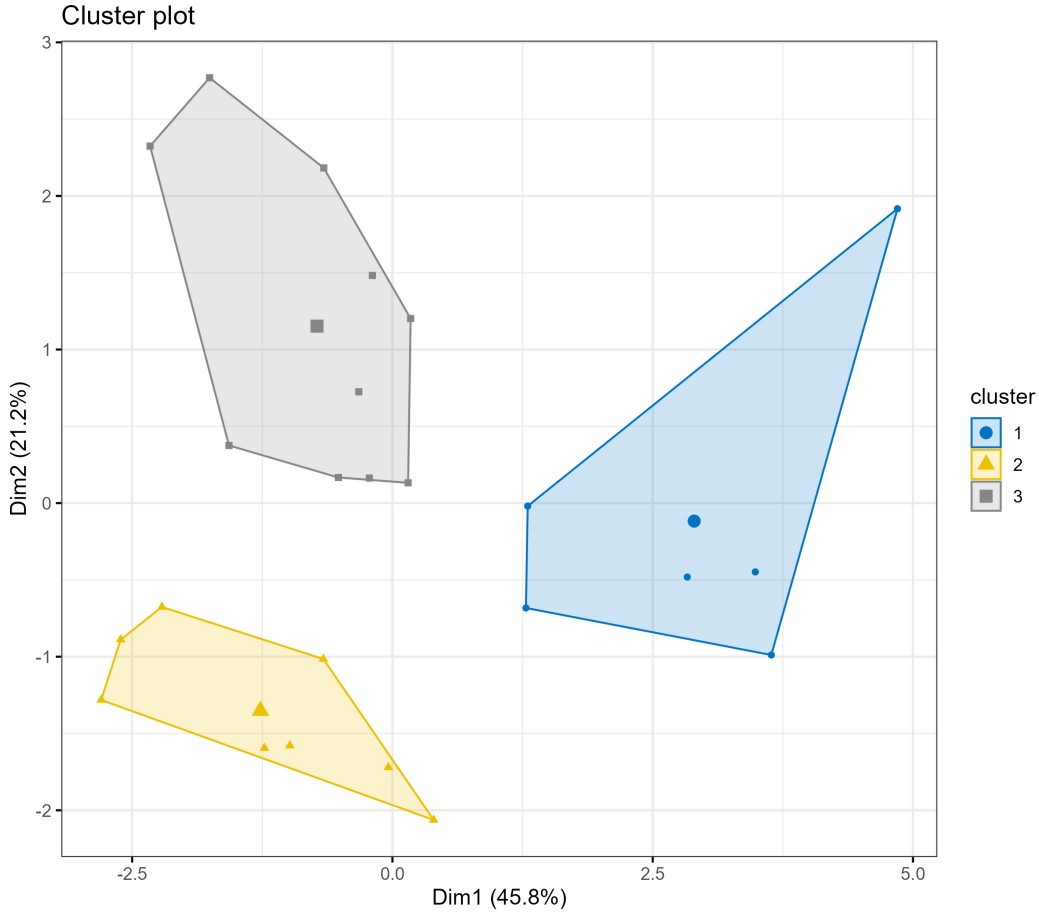

**Figure 4   Athlete clustering based on baseline characteristics.**

## PBT: a more potent stimulus for power and skill transfer

Our main analysis revealed significant Group × Time interactions favoring PBRT in multiple explosive tasks, including four kg medicine ball throws and, most notably, power output in the sport-specific Seoi-nage (Table 6, Fig. 3). The question then arises: why did the supposedly more "advanced" and "individualized" VBRT protocol underperform compared to the traditional PBRT method?

Several converging mechanisms may explain this outcome. First, the fundamental driver of long-term strength and power adaptation is the mechanical and metabolic stress placed upon the neuromuscular system. PBRT, by its very nature, enforces a high level of mechanical tension through a fixed, challenging load, which is a primary catalyst for strength adaptation (*Schoenfeld, 2010*). Furthermore, the requirement to complete all prescribed repetitions, even as fatigue causes movement velocity to slow (*i.e.,* "grinding reps"), ensures that athletes achieve or approach momentary muscular failure. This process is crucial for maximizing the recruitment of high-threshold motor units, which are essential for force production and power expression (*Willardson, 2007*). VBRT, conversely, may preemptively curtail this vital stimulus. By instructing athletes to maintain a certain velocity,

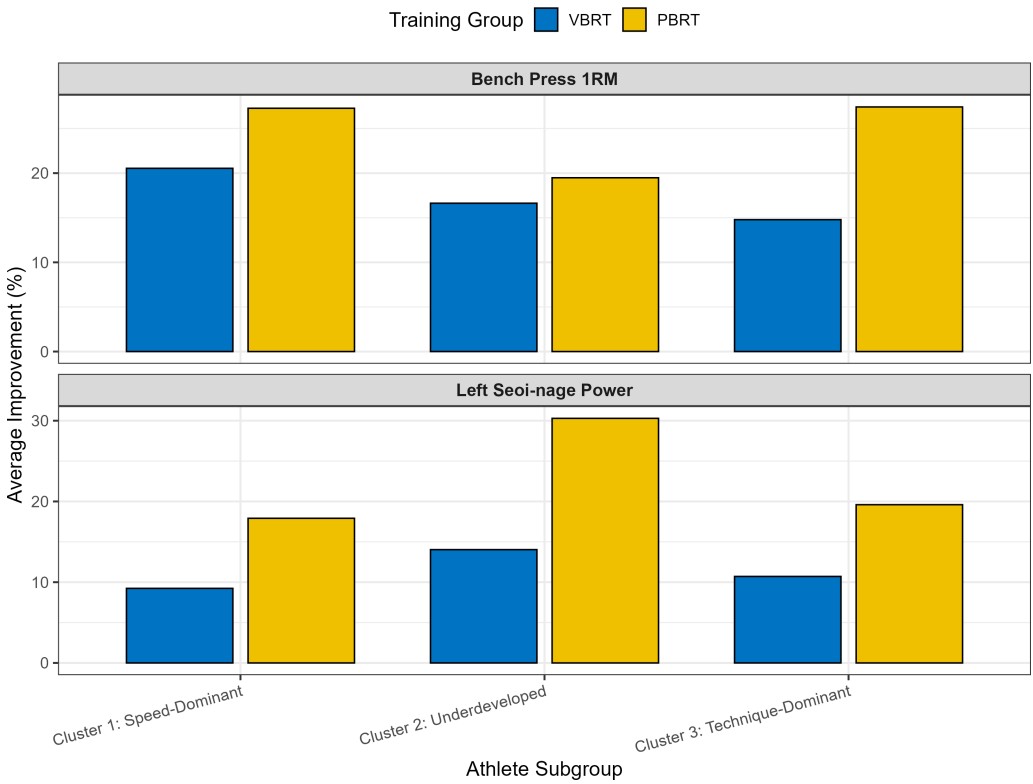

**Figure 5** Training response by athlete profile.

and by automatically reducing the load when this is not possible, the VBRT protocol may have prevented our athletes from pushing through these challenging but highly adaptive repetitions.

Second, this leads to the most critical, albeit confounding, factor: training volume-load. While not directly equated in this study—a significant limitation—it is highly plausible that the PBRT group accumulated a greater total volume-load over the 8 weeks. The dose–response relationship between training volume and muscular adaptation is well-established (*Schoenfeld, Ogborn & Krieger, 2017*). If the VBRT protocol's auto-regulation consistently led to lighter training loads to preserve velocity, it may have simply resulted in an insufficient overall training dose to drive superior adaptations. This suggests a potential paradox in VBRT application: in its effort to manage fatigue on a micro-level (within-session), it may compromise the macro-level stimulus necessary for long-term progress.

Finally, the superior transfer of training to the Seoi-nage task in the PBRT group is particularly noteworthy. This suggests that the adaptations from PBRT were more specific to the demands of a dynamic, full-body throw against resistance. This could be explained by the principle of maximal volitional intent. Seminal research has demonstrated that the intent to move a resistance as explosively as possible, rather than the actual velocity achieved, is a key determinant of training outcomes (*Behm & Sale, 1993*; *Behm et al., 2024*). PBRT inherently forces maximal intent against a consistently heavy load, a condition that

may better replicate the neuromuscular requirements of executing a powerful throw on a resisting opponent than lifting a lighter, auto-adjusted weight at a higher velocity.

## Individualization revisited: pbrt's robustness across athlete profiles

A key aim of our study was to explore individualized responses. Our exploratory cluster analysis, an approach conceptually aligned with the growing trend of using athlete profiles to guide personalized training (*Jiménez-Reyes et al., 2017*; *Jiménez-Reyes, Samozino & Morin, 2019*), yielded a surprising result. We identified three distinct athlete profiles: "Speed-Dominant", "Underdeveloped", and "Technique-Dominant" (Fig. 4).

One might hypothesize that "Speed-Dominant" athletes would thrive under a VBRT protocol. However, our data showed the opposite. As seen in Fig. 5, PBRT elicited superior or comparable improvements in strength and sport-specific power across all three subgroups. This robust finding suggests that, at least for this cohort and training phase, the benefits of the consistent, high-effort stimulus provided by PBRT transcended baseline individual differences. The "individualized" approach of VBRT did not prove superior for any identified subgroup; instead, the "one-size" approach of PBRT was universally more effective. This does not negate the value of individualization, but rather suggests that for foundational power development, ensuring a sufficient and consistent training stimulus may be a more critical factor than real-time velocity modulation.

## Limitations and future research directions

The most significant limitation of this study is the lack of equated training volume-load. This is a well-documented challenge in the literature comparing auto-regulated and traditional training paradigms (*Orange et al., 2022*). Indeed, recent meta-analyses finding trivial differences between VBT and traditional methods often cite this methodological inconsistency as a major issue (*Orange et al., 2022*; *Liao et al., 2021*). Future research must prioritize study designs where total work is matched to truly isolate the effect of velocity feedback.

Additionally, our sample size, while adequate for the primary ANOVA, was small for the exploratory cluster analysis. The athlete profiles identified should be considered preliminary and require validation in larger, more diverse cohorts. It is also important to acknowledge that our models did not account for uncontrolled variables such as athlete genetics, nutrition, and sleep quality, which are known to influence training outcomes (*Halson, 2014*; *Mason et al., 2023*). Finally, the use of a single upper-body exercise limits the generalizability of these findings to whole-body or lower-body dominant movements.

## Practical applications

Based on our findings, coaches working with combat sports athletes should consider the following:

- **Reaffirm the value of PBRT:** for developing foundational strength and improving its transfer to explosive and sport-specific power, PBRT remains a highly effective and reliable methodology, especially during off-season or preparatory phases.
- **Implement VBRT with a clear purpose and caution:** VBRT is an invaluable tool for objective monitoring, providing immediate feedback, and managing fatigue. However,

coaches should be cautious that its auto-regulatory nature does not lead to a systematic reduction in the necessary training stimulus. It may be most appropriate for in-season maintenance, peaking phases where movement quality is paramount, or for athletes known to be highly susceptible to overtraining.

- **Consider a periodized, hybrid approach:** an optimal long-term program might integrate both methods. For example, using PBRT to build a strong foundation, followed by a block of VBRT to refine power expression and manage fatigue closer to competition.

## CONCLUSION

In conclusion, this study found that for the cohort of university-level combat sports athletes investigated, an 8-week program of traditional percentage-based resistance training (PBRT) was significantly more effective than velocity-based resistance training (VBRT) at improving measures of general upper-body power and sport-specific power. This superiority of PBRT was observed across various medicine ball throw metrics and in the power output of a complex, sport-specific throwing technique. Furthermore, exploratory analysis revealed that this advantage was consistent across athletes with different baseline profiles, including those who were speed-dominant, underdeveloped, or technique-dominant. These results challenge the assumption of VBRT's universal advantage for power development and underscore the continued importance and efficacy of structured, high-effort PBRT in strength and conditioning programs for combat sports. While VBRT remains a valuable tool for monitoring and fatigue management, its implementation must be carefully managed to ensure an adequate training stimulus is delivered to drive meaningful adaptation.

### Funding
This research was supported by the Guangdong Provincial Philosophy and Social Sciences Regularization Project 2022 (GD22CTY09): Research on the Coordinated Development Path of International Competitiveness in Sports in the Guangdong-Hong Kong-Macao Greater Bay Area. The funders had no role in study design, data collection and analysis, decision to publish, or preparation of the manuscript.

### Grant Disclosures
The following grant information was disclosed by the authors:
Guangdong Provincial Philosophy and Social Sciences Regularization Project 2022: GD22CTY09.

### Competing Interests
The authors declare there are no competing interests.

### Author Contributions

- JiaYong Chen conceived and designed the experiments, performed the experiments, authored or reviewed drafts of the article, and approved the final draft.

- Beiwang Deng performed the experiments, prepared figures and/or tables, and approved the final draft.
- Tianyuan He analyzed the data, prepared figures and/or tables, and approved the final draft.
- Jiaxin He analyzed the data, authored or reviewed drafts of the article, and approved the final draft.
- Duanying Li conceived and designed the experiments, authored or reviewed drafts of the article, and approved the final draft.
- Min Lu analyzed the data, authored or reviewed drafts of the article, and approved the final draft.
- Jian Sun analyzed the data, authored or reviewed drafts of the article, and approved the final draft.

## Human Ethics

The following information was supplied relating to ethical approvals (i.e., approving body and any reference numbers):

The University of Guangzhou sport Univeisity Ethical approval to perform the study within its facilities (Ethical Application Ref: 2023LCLL-73).

## Data Availability

Raw data is available in the Supplemental Files.

## Supplemental Information

Supplemental information for this article can be found online at http://dx.doi.org/10.7717/peerj.19928#supplemental-information.

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
