# Peer review of "Individualized responses to velocity-based versus percentage-based resistance training in combat sports athletes: the influence of baseline characteristics"

_PeerJ, doi:10.7717/peerj.19928_

## Round 0.1 · original submission · Major Revisions

**Language Note:** PeerJ staff have identified that the English language needs to be improved. When you prepare your next revision, please either (i) have a colleague who is proficient in English and familiar with the subject matter review your manuscript, or (ii) contact a professional editing service to review your manuscript. PeerJ can provide language editing services - you can contact us at [email protected] for pricing (be sure to provide your manuscript number and title). – PeerJ Staff

·

Basic reporting

Overall Impression:
The manuscript investigates the effects of velocity-based resistance training (VBRT) versus percentage-based resistance training (PBRT) on upper limb strength and power in combat sports athletes, with an additional focus on individualised responses. The study addresses a pertinent research question in applied sports science and employs a randomised parallel-group controlled trial. The authors have commendably attempted to explore nuanced aspects such as the influence of baseline levels and individual response patterns. However, some areas require further clarification and revision to enhance the manuscript's scientific rigour, clarity, and interpretability, thereby meeting the standards for publication. Key areas for improvement include the specificity of the title, justification for certain methodological choices (particularly the primary training exercise and assessment of participant experience), the detail and consistency of reporting (especially regarding participant characteristics in the abstract and statistical results in the discussion), the consistent visual indication of statistical significance in tables and figures, and a more thorough consideration and potential analytical address of the implications of non-equivalent training doses between groups.

- Clear and unambiguous, professional English used throughout: The manuscript is generally written in clear English. However, minor refinements in phrasing could enhance precision in places.
- Literature references, sufficient field background/context provided: The introduction (lines 38-83) adequately establishes the importance of upper limb strength and power in combat sports and outlines the rationale for comparing PBRT and VBRT. Relevant prior literature is generally cited. However, the discussion of existing VBRT research specifically within combat sports could be more comprehensive, building on the authors' note (lines 65-67: "current research on VBRT for combat sports athletes such as judo and wrestling remains limited, and conclusions across different studies are inconsistent , necessitating further validation of its effectiveness").
- Professional article structure, figures, tables. Raw data shared:
Structure: The article adheres to a standard scientific structure (Abstract, Introduction, Methods, Results, Discussion, Conclusion).

Title: The current title, "Velocity-based resistance training vs. percentage-based resistance training: The best approach is individualized," while highlighting a key theme, is rather broad and reads more like a general conclusion than a specific descriptor of this study. The authors should consider revising the title to more accurately reflect the specific investigation undertaken, particularly its focus on individualised responses within their specific cohort and the influence of baseline characteristics in combat sports athletes. For instance, a title such as 'Individualised Responses to Velocity-Based versus Percentage-Based Resistance Training in Combat Sports Athletes' or 'Baseline Strength Influences Adaptations: Comparing Velocity-Based and Percentage-Based Resistance Training in Combat Sports Athletes' would better encapsulate the study's scope and nuanced findings.

Abstract (Lines 2-34): The abstract provides a reasonable overview. However, to provide immediate context for the reader, it should include key participant characteristics (e.g., age, general training experience, and perhaps baseline bench press 1RM, presented as mean ± SD). These data are available in Table 1 of the manuscript.

- Tables and Figures:
Clarity and Indication of Significance: While p-values are generally reported in tables, the authors should ensure that statistical significance is consistently and clearly indicated in all relevant tables and figures where comparisons are made or effects are reported (e.g., using asterisks for p-value thresholds, or significance bars in figures). For instance, Table 10 appropriately uses asterisks; this practice should be extended to other tables presenting significance tests (e.g., Tables 6, 7, 8, 9) and incorporated into figures (e.g., Figures 3, 4, 5, 6) to facilitate rapid interpretation of results.

Table Labelling/Content Mismatch: There appears to be a discrepancy in the manuscript body concerning Table 3 and Table 4. The text under section 2.3.2 "Test Indicators" (lines 139-146) describes the training intervention details. However, Table 3 (as per peerj-116413-Table_3.docx) is correctly titled "Test Indicators" and lists outcome measures. Conversely, section 2.4.1 "Overview of Training Protocol" (lines 150-157) also describes the training intervention, referencing Table 4 "Experimental Protocol". This redundancy and mismatch between section heading 2.3.2 and its textual content needs correction. The text in lines 139-146 should accurately describe Table 3, or be moved to section 2.4.1.

Figure 6 (Comparison of relative change rates): For the box plots with overlaid individual data points, ensure y-axis labels are explicit (e.g., "Relative Change Rate (%)") and that any visual representation of statistical comparison is clear.

Self-contained with relevant results to hypotheses: The study presents a self-contained body of work relevant to its stated hypotheses.

Experimental design

Original primary research within Aims and Scope of the journal: The research is original primary research and aligns with the Aims and Scope of journals publishing sports science and strength and conditioning studies.

Research question well defined, relevant & meaningful. It is stated how research fills an identified knowledge gap: The research questions are clearly defined (lines 70-76) and address relevant issues in training combat athletes. The study aims to fill identified knowledge gaps, such as the impact of baseline levels (lines 67-69).

Rigorous investigation performed to a high technical & ethical standard:
Ethical approval (2023LCLL-73) and informed consent are reported (lines 121-126).

Exercise Selection (Bench Press): The manuscript should provide a more robust justification for selecting the bench press as the sole resistance training exercise for judo and wrestling athletes. Specifically, discuss its direct transferability to the complex, multi-planar demands of grappling techniques (e.g., Seoi-nage), beyond its role as a general upper limb strength developer, perhaps drawing further on the literature cited in lines 41-43.

Participant Characteristics (Resistance Training/Bench Press Experience): Table 1 details general "Training Experience (years)". However, specific details regarding participants' prior resistance training history, and crucially, their experience and proficiency with the bench press exercise itself, are absent. This is a significant omission as it could substantially influence baseline values and training adaptations.

Homogeneity and Specific Combat Sport: Line 111 states participants had "training levels as homogeneous as possible". Clarification on how homogeneity was assessed is needed. The distribution of wrestling versus judoka athletes within each training group should also be provided.

VBRT Protocol Details (Velocity Threshold): The rationale for the ±0.06m/s velocity deviation threshold for load adjustment (line 166) should be briefly explained within the text, linking it to the intended training stimulus.

Standardisation (Recovery): Details on how "recovery and regeneration relaxation" exercises (lines 155-156) were standardised should be provided.

Methods described with sufficient detail & information to replicate: The methods are generally well-described. Addressing the above points above would enhance replicability.

Validity of the findings

All underlying data have been provided; they are robust, statistically sound, & controlled:
The statistical approach appears generally appropriate (lines 178-208).

Reporting Inconsistencies/Accuracy:
In the Discussion (lines 362-368), the statement regarding significant VBRT superiority for "6kg medicine ball throws" (power and distance) appears inconsistent with data in Table 8 (p = 0.267 for power, p = 0.265 for distance) unless I'm mistaken.

In Table 8, for "4kg Medicine Ball Velocity," the "t/W Value" of "15" with "p Value" of "0.000" lacks clarity. The nature of the test statistic (t or W) should be clear, and the precise p-value reported (e.g., p < 0.001 if not exactly zero).

Training Dose Comparability (Key Limitation): The authors rightly acknowledge the "insufficient control of training protocol dosage equivalence" (lines 450-453) as a key confounding factor. This makes it challenging to attribute differences purely to the training methodology versus potential differences in the actual training stimulus.

Conclusions are well stated, linked to the original research question & limited to supporting results:

Conclusions generally reflect the findings. However, claims of superiority (e.g., Abstract, lines 32-33, regarding bench press 1RM) must be carefully qualified by the training dose limitation and the specific analytical approach (e.g., regression considering baseline).

Additional comments

The manuscript presents a valuable investigation into VBRT and PBRT, with a notable attempt to explore individual responses. Key areas for improvement include greater specificity in methodological justifications, enhanced detail in participant characterisation, meticulous consistency in results reporting (including visual indication of significance), and a deeper consideration of the non-equivalent training dose limitation.

·

Basic reporting

The topic of the study is well-chosen and relevant. However, the sample size appears to be insufficient to draw robust conclusions.
What is the distinction between VBRT and VBT in the study? Aren’t these methods generally encompassed under the umbrella term “VBT”? This point requires clarification.
The statistical analysis method needs to be revised. The current approach may not be appropriate given the study design.

Experimental design

The mean age of the participants should be included in the abstract (such as: mean: 22.22; SD: 1.11…).
It is known that 1RM values may vary on different days (Shattock, Kevin. A comparison of velocity based training versus Rate of Perceived Exertion / Repetitions in Reserve based intensity methodology on measures of strength, power & speed in men’s senior rugby union, 2018). How was this issue addressed in the measurement procedures?
What is the difference between VBRT and VBT in the study? This distinction should be clearly emphasized in the significance of the study.
The introduction is too long and should be shortened.

Validity of the findings

A sample size of 12 participants is insufficient. The reported sample size of 16 – does it refer to one group or the total number? How was development observed over the eight-week period?
Why were 4–6 kg loads selected? What is their relevance to the participants’ sport discipline?
The use of the R software is appropriate, but which statistical package was used for analysis?
Repeated Measures ANOVA should be applied in this study, as the pre- and post-test measurements are dependent. Using t-tests or Mann-Whitney tests may increase the risk of Type I error. The study should examine pre- and post-test values. However, regression analysis has been used here to assess the effect.

Additional comments

Line 432: “athlete genetics, nutrition, sleep, and psychological” – How were these factors controlled or excluded in the present study?

Lines 435–448: This section should be part of the Conclusion. If it is to be included in the Discussion, proper referencing is necessary.

The Discussion section needs to be restructured.

---

## Round 0.2 · accepted · Accept

Thank you for addressing the previous reviewer comments, both of whom were satisfied with the changes made to your submission. Accordingly, I am pleased to accept your manuscript for publication.

Upon reviewing your work, I did notice some minor writing and presentation errors throughout, so please ensure these are rectified during the proofing stage when the article is prepared for publication in the journal. For instance, there are some writing errors and incorrect terminology used in places (e.g., "weight" instead of "body mass"), there is presentation of bullet points in text when the detail should be presented in paragraph form (e.g., the clusters in the results section), and the presentation of tables (e.g., Tables 5 and 6) appears to be in output form instead of being constructed as new tables for publication. Congratulations once again on your research!

·

Basic reporting

The authors addressed all feedback by clarifying the title and abstract, defining participant characteristics, and tightening language throughout. Figures and tables were recreated for consistency and clarity, and all data and labels are now clear. These improvements have made the manuscript fully transparent and self-contained.

Experimental design

In response to suggestions, the authors justified exercise selection, detailed participant experience, and explained the randomisation and group balance methods. Protocols and statistical approaches are now clear and replicable, and ethics are fully documented, strengthening the study’s rigour.

Validity of the findings

All concerns about data reporting and statistical interpretation have been resolved. The authors clarified analyses, explained statistical choices, and ensured that results are consistent across text, tables, and figures. The expanded discussion on training dose strengthens interpretation and appropriately limits conclusions.

Additional comments

All prior issues have been addressed in full. The revisions significantly improve clarity, transparency, and scientific quality.

·

Basic reporting

The author(s) made the corrections I requested. The topic of the study is understandable, and it is a highly original paper.

Experimental design

Statistical analysis corrected.

Validity of the findings

The results were discussed in the discussion section.